# Recent Advances in Pancreatic Cancer: Novel Prognostic Biomarkers and Targeted Therapy—A Review of the Literature

**DOI:** 10.3390/biom11101469

**Published:** 2021-10-06

**Authors:** Konstantin Schlick, Dominik Kiem, Richard Greil

**Affiliations:** 1Oncologic Center, Department of Internal Medicine III with Haematology, Medical Oncology, Haemostaseology, Infectiology and Rheumatology, Paracelsus Medical University, 5020 Salzburg, Austria; k.schlick@salk.at (K.S.); d.kiem@salk.at (D.K.); 2Cancer Cluster Salzburg, 5020 Salzburg, Austria; 3Laboratory for Immunological and Molecular Cancer Research (SCRI-LIMCR), Salzburg Cancer Research Institute, 5020 Salzburg, Austria

**Keywords:** pancreatic cancer, miRNAs, chemoresistance, targeted therapy, immunotherapy

## Abstract

Pancreatic adenocarcinoma carries a devastating prognosis. For locally advanced and metastatic disease, several chemotherapeutic regimens are currently being used. Over the past years, novel approaches have included targeting EGFR, NTRK, PARP, K-Ras as well as stroma and fibrosis, leading to approval of NTRK and PARP inhibitors. Moreover, immune check point inhibitors and different combinational approaches involving immunotherapeutic agents are being investigated in many clinical trials. MiRNAs represent a novel tool and are thought to greatly improve management by allowing for earlier diagnosis and for more precise guidance of treatment.

## 1. Introduction

For many years, chemotherapeutic regimens have been the cornerstone of therapy for inoperable pancreatic adenocarcinoma (PC) (i.e., locally advanced or metastatic) and evolved as adjuvant or neoadjuvant treatment for resectable pancreatic carcinoma [1,2]. Overall, however, pancreatic carcinoma shows only moderate sensitivity to chemotherapy [3]. This is reflected by the overall dismal prognosis, that has not improved over the last years [4]. In the following review of the available literature, we would like to provide a summary of novel approaches that specifically aim at targeting the malignant cells and their surroundings, i.e., the microenvironment and the stroma. We will discuss some approaches over the last years that failed and how the positive results for the PARP inhibitor olaparib in the POLO trial finally allow for targeted therapy to become clinically useful in PC [5,6]. Other well-investigated targets are EGFR, K-RAS, and more recently, NTRK 8 [2,7,8]. As extensive fibrosis is a prominent feature of PC, some of the most interesting approaches at targeting it are included in this review. Immunotherapy has revolutionized the treatment of many solid tumor entities, starting with melanoma, and expanding further since then. PC, however, is amongst the less responsive entities and so far, immunotherapy has not been successful. We summarize novel immunotherapeutic approaches with a focus on combinations of several agents.

In the 2nd part of our review we are going to discuss micro-RNAs as possible novel biomarkers and their prognostic and predictive roles in pancreatic cancer. MicroRNAs are small, approximately 22 nucleotides long non-coding single-stranded RNAs, regulating gene expression at a posttranscriptional level. MicroRNAs act as tumor suppressors by negatively regulating oncogenes, i.e., genes that promote aberrant cell proliferation, and thereby inhibit cell division [9]. Currently, microRNAs are of interest in various cancer entities for their diagnostic, prognostic and predictive roles as biomarkers.

In PC, a distinct microRNA expression profile compared to benign lesions has been observed [10]. By using microRNA expression signatures, a clear discrimination between healthy, inflamed and cancerous pancreatic tissue can be made [10,11,12]. These findings are of clinical relevance, because histological clarification of pancreatic lesions is often challenging, particularly in tissues obtained by endoscopic ultrasound-assisted fine-needle biopsies. Furthermore, an increasing number of publications in recent years correlate miRNA expression in PC with resistance or sensitivity towards various chemotherapeutic agents.

## 2. Targeted Therapy

The following section will describe targeted therapy in depth. A summary of all the clinical trials discusses in Section 2 and Section 3 is provided in Appendix A. Figure 1 shows important (immuno-)therapeutic targets in PC.

### 2.1. Targeting EGFR

Discussion should start with the first approved targeted therapy for PC, the epidermal growth factor receptor (EGFR) tyrosine kinase inhibitor erlotinib. In the early 2000s, a phase III trial showed a statistically significant prolongation of overall survival (OS) by addition of erlotinib to gemcitabine (6.24 vs. 5.91 months) [13]. Arguably, this difference is too small to be clinically relevant and given the side effects of erlotinib that mainly affect the skin, erlotinib is used in routine clinical practice. A later phase III trial investigating the blockade of EGFR signaling by the monoclonal antibody cetuximab with gemcitabine did not result in any benefit, and another phase III trial exploring gemcitabine or capecitabine with erlotinib has reported negative results, too [14,15]. As a side note, addition of erlotinib to gemcitabine did not show any benefit in OS in the adjuvant setting [16].

### 2.2. Targeting NTRK

Recently, larotrectinib and entrectinib, inhibitors of the neurotrophic tyrosine receptor kinase (NTRK) fusion genes, have received approval in many countries for tumors harboring the mutation, independently of their origin [17,18,19]. Their gene products, tropomyosin receptor kinases, have been known protooncogenes for over 20 years due to their capability to induce signals of proliferation via activation of MAPK, PI3K, and PKC pathways [8]. A small fraction of patients with PC harbors this mutation (<5%) [8]. For larotrectinib, a pooled analysis of three phase 1 and 2 trials (NCT02122913, NCT02637687, NCT02576431) included 159 patients, two of them with PC, of which one had a positive response [20]. With an overall response rate of 79% for all tumor types, larotrectinib may become a valid treatment option for a subgroup of patients with PC. Similarly, an analysis of three phase 1 and 2 clinical trials for entrectinib (ALKA-372–001, STARTRK-1, STARTRK-2) analyzed 54 patients, three of which with PC [7]. The overall response rate was close to 60%.

### 2.3. Targeting PARP

Attention has also been brought to the poly(adenosine diphosphate–ribose) polymerase (PARP) inhibitors. Prevention of PARP to repair single-strand breaks eventually leads to accumulation of double strand breaks, to which cells with non-intact double strand repair, notably because of BCRA1/2 mutation, are sensitive [21]. In addition to being efficacious in BRCA1/2 positive ovarian carcinoma and breast cancer, olaparib has been successfully investigated for pretreated pancreatic carcinoma [22]. After a phase III trial has shown an increase in the median progression-free overall surival (PFS) from 3.8 to 7.4 months for patients receiving four cycles of olaparib as maintenance therapy compared to best supportive care after a platinum-based first line therapy, it has been approved in the U.S. for this indication [6]. Overall survival, however, might not be influenced by the administration of olaparib [5]. Several phase I and II clinical trials have investigated another PARP inhibitor, veliparib. In 2018, a phase II trial reported stable disease in 25% of patients for veliparib monotherapy [23]. Following this, veliparib was investigated in combination with gemcitabine and radiotherapy for locally advanced PC (phase 1) and a benefit in median OS of 5 months (19 versus 14 months) was reported for patients harboring mutations affecting DNA repair mechanisms as compared to individuals with wild-type status of DNA damage repair genes [24]. Veliparib with 5-FU and oxaliplatin for metastatic disease (phase I/II, single-arm trial) showed an overall response rate (ORR) of 57% for platinum-naïve patients with these mutations [25]. Another phase II trial, however, did not observe any improvement in response rates for the addition of veliparib to gemcitabine plus cisplatin [26]. Further phase III clinical trials are thus needed to clarify the role of PARP inhibitors in clinical management.

### 2.4. Targeting K-RAs

In approximately 95% of PC patients, an activating mutation in K-Ras is found and plays a central role in initiation, maintenance, and progression of disease [27]. For a long time, approaches to target Ras have failed and Ras was therefore considered to be undruggable [28]. Around a decade ago, adjuvant treatment with K-Ras vaccines have brought rather discouraging results in resectable PC [29,30]. One study observed an immunological response to vaccination, but the median OS in did not significantly increase for responders and the 5-year survival rate of 20% of the entire cohort increased to 24% for responders [29]. In another trial, no immunological response was induced by the vaccine [30]. In PC, also clinical trials to target downstream pathways of K-Ras have failed so far [27,31]. Recently, this paradigm has been changing. The development of the K-Ras inhibitor sotorasib demonstrated the possibility of directly targeting K-Ras (although effective only for the G12C mutation, not the much more common G12D in pancreatic carcinoma) [32]. The mDC3/8 dendritic cell vaccine designed to target mutant K-Ras is currently investigated in a phase I trial (NCT03592888).

### 2.5. Targeting the Stroma, Fibrosis, and Extracellular Matrix

The main factors leading to chemoresistance are to be found in the cancer microenvironment and the extensive fibrosis surrounding the tumor, as reviewed in depth by Schober and colleagues [33]. Naturally, attempts to target the cancer microenvironment have been made. More than a decade ago, an approach to increase drug delivery into the tumor included successful experiments with inhibition of hedgehog signaling in mice [34]. Subsequent clinical trials, however, failed to show any benefit [35,36,37] This includes a randomized phase Ib/II trial, where addition of vismogedib or placebo to gemcitabine did not alter OS [35]. For gemcitabine in combination with saridegib (another hedgehog inhibitor), a decreased OS compared to gemcitabine plus placebo at interim analysis led to a premature end of the study and subsequently also put a stop to a clinical trial of FOLFIRINOX and saridegib [35,36]. In a more recent phase II study, the addition of vismogedib to gemcitabine plus nab-paclitaxel as first-line therapy resulted in a median OS of 5.4 months, discouraging further research, as the median OS for the chemotherapeutic regimen alone has been reported to be equal or higher [37,38].

In a similar approach to facilitate chemotherapeutic penetration into tumor tissue, enzymatic lysis of extracellular hyaluronic acid by PEGPH20 plus gemcitabine was promising in vitro and in vivo [39]. Investigations carried on, until the HALO 109-301 phase III trial did not show any benefit in OS and PFS [40].

TGF-beta is a central cytokine in the tumorigenesis of PC, as well as its microenvironment, and three-dimensional in-vitro models for PC underline a direct relation between TGF-beta and the architecture [41,42,43]. TGF-beta has recently become of interest as a therapeutic target: Preclinical experiments demonstrated a reduction of the tumor cells’ aggressiveness, if secretion of TGF-beta in pancreatic stellate was abrogated [44]. Another interesting aspect lies in the fact that cancer-associated fibroblasts (CAFs), which are central for the desmoplastic stroma, rely on TGF-beta signaling and therefore, inhibition of TGF-beta has been explored as one strategy to target CAFs [45]. A phase Ib/II clinical trial in patients with unresectable PC reported an increase in OS from 7.1 to 8.9 months for the addition of the TGF-beta inhibitor galunisertib to gemcitabine [46].

Another attempt of stromal remodeling to increase chemotherapeutic delivery took advantage of the overexpression of the vitamin D receptor, which according to a mice model, serves as a transcriptional master regulator of the tumor stroma in this tumor entity, mainly by affecting PSCs [47]. As preclinical data were promising, many clinical trials have been started, more recently combining immune checkpoint therapy as well as oncolytic vaccine therapy with vitamin D analogues [48].

As PSCs also overexpress the vitamin A receptor, and its inhibition decreases the deposition of extracellular matrix, a phase II clinical trial is currently ongoing to explore the combination of gemcitabine/nab-paclitaxel and all-trans retinoic acid in patients with locally advanced and metastatic disease (NCT03307148) [49,50].

Inhibition of neoangiogenesis has been investigated but targeting VEGF with a TKI (axitinib) or a monoclonal antibody (bevacizumab) plus gemcitabine reported negative results in terms of no benefit in overall surivival [51,52].

## 3. Combined Immunotherapies

In addition to fibrosis, the immunosuppressive properties of the microenvironment are considered crucial for disease progression and resistance to therapy [53,54,55]. It consists of a complex network of different immune cell populations, including myeloid-derived suppressor cells, macrophages, and regulatory T-cells, but not cytotoxic T-cells. As these are thought to exert their effect by expression of PD-1, CTLA-4, CD-40 as well as secretion of TGF-beta, many novel approaches have aimed at targeting these proteins, some of which we will discuss in the following paragraphs [54].

### 3.1. Targeting PD-L1/PD-1

Tumors characterized by high microsatellite instability (MSI-high) or mismatch-repair deficiency (dMMR) are susceptible to immune checkpoint therapy [56]. It is thought that the increased neoantigen expression by the tumor cells is recognized by the immune system [57,58]. In the phase II KEYNOTE-158 study investigating the PD-1 inhibitor pembrolizumab for dMMR/MSI-high tumors, 9.4% of the patients included suffered from PC, making it the third most included tumor entity. With an ORR of 18% and a PFS of 2.2 months, however, it was among the least responsive tumors, and of the six most included types of tumors, only brain tumors with an overall response rate of 0% performed worse [59]. Based on this trial, pembrolizumab has received FDA approval for MSI-high metastatic tumors for pretreated patients with no effective other line of treatment. For pancreatic carcinoma, several attempts at targeting PD-L1/PD-1 as part of a combination are ongoing.

The combination of pembrolizumab with the CXCR4 antagonist motixaforatide (and chemotherapy) has been investigated in the COMBAT trial (phase IIa) with promising results especially in the arm with additional chemotherapy: A disease control rate (DCR) of 32% was reported for motixaforatide plus pembrolizumab and a DCR of 77% and ORR of 32% with additional chemotherapy, encouraging further randomized trials [60]. Also, an increased cytotoxic T-cell infiltration into the tumor and a decrease in immunosuppressive cells were observed.

Another attempt to increase the efficacy of pembrolizumab by increasing T-cell infiltration and PD-L1 expression yielded into a phase Ib trial that combined pembrolizmab with chemotherapy and the oncolytic virus pelareorep. Due to promising results, a phase II trial (NCT03723915) is now ongoing [61]. The abovementioned PEGPH20 is currently being evaluated in combination with pembrolizumab as part of a phase II trial (NCT03634332). As recently reviewed by Arias-Pinilla and Modjtahedi, several other early phase trials investigating pembrolizumab or nivolumab in combination with olaparib, with paricalcitol plus gemcitabine and nab-paclitaxel, with a chemokine receptor agonist for CCR2/5, or with a cytokine receptor antagonist for CXCR1/2 are ongoing [62].

Two other ongoing trials involve the novel PD-1 inhibitor spartalizumab. For the combination of spartalizumab with the anti-IL-6 monoclonal antibody siltuximab, a phase Ib/II trial is ongoing (NCT04191421). Another phase I trial is investigating spartalizumab with nab-paclitaxel, gemcitabine, and the anti Il-1 beta monoclonal antibody canakinumab (NCT04581343) [62].

For the combination of the anti-PD-L1 monoclonal antibody durvalumab with the anti-CTLA-4 antibody tremelimumab, a phase II clinical trial reported a discouraging objective response rate of 3.1% only (and 0% for durvalumab monotherapy) [63].

### 3.2. Targeting CTLA-4

Ipilimumab, an inhibitory monoclonal antibody targeting CTLA-4, has been unsuccessfully investigated in a phase II clinical trial in advanced PC [64]. The combination of ipilimumab with the GM-CSF cell-based vaccines (GVAX), however, showed superior activity in a later trial, encouraging further studies [65]. Taking advantage of the T-cell priming by GVAX, an additional boost with the listeria monocytogenes-expressing mesothelin vaccine CRS-207 significantly increased overall survival from 4.6 to 9.7 months [66]. Therefore, the combination of GVAX and CRS-207 was evaluated together with ipilimumab, where, unfortunately, the addition CTLA-4 immune checkpoint inhibition did not prolong survival [67]. Other recent studies investigated the addition of ipilimumab to gemcitabine without increasing efficacy, or reported inferiority of ipilimumab with GVAX compared to continued administration of FOLFIRNOX [68,69]. Four ongoing trials investigate the combination of ipilimumab with nivolumab in PC (NCT04361162, NCT04258150, NCT03104439, NCT04247165).

### 3.3. Targeting CD40

Already in 2011, a study involving mice models and human tissue demonstrated that CD40 agonists could aid at tumor regression in patients with PC by activating macrophages to fight the tumor and by leading to regression of the stroma [70]. In combination with chemotherapeutic agents, CD40 agonists have also been shown to help overcome resistance to immune checkpoint inhibitors in mice models [71]. More recently, a proof-of concept study in a murine model showed that although CD40-agonist monotherapy did not improve OS, it increased susceptibility of PC to subsequent dendritic cell vaccinations, arguably by unleashing a T-cell response, and resulted in prolongation of the OS [72]. Despite a phase I trial investigating the combination of CP-870,893 (a CD40 monoclonal antibody) with gemcitabine showed antitumor activity, no further trials have been conducted so far [73].

### 3.4. Targeting IL-10

Another attempt to increase T-cell infiltration into the tumor took advantage of recombinant Il-10 (pegilodecakin) [74]. Whereas preclinical and early clinical data were promising, the phase III SEQUIOA trial failed to show a benefit in OS (5.8 vs. 6.3 months) and PFS (2.1 months both groups) for the addition of pegilodecakin to FOLFOX [75,76]. For an overview of selected studies discussed in the text, please refer to Appendix A.

## 4. miRNA

Several efforts have been performed during the last years to enable physicians to diagnose cancer at an earlier stage of disease and to identify biomarkers to guide treatment. As pancreatic cancer remains asymptomatic over a long period of time, diagnoses at an early stage of disease is possible in 15–20% of patients only. Only these patients are potential candidates for curative surgical resection, resulting in a median survival of up to 24 months, if a R0 resection is achieved [77,78]. Micro RNAs (miRNAs) have been propeosed as biomarkers potentially allowing for earlier diagnosis and will be discussed in the following section. A summary of miRNAs can be found in Appendix A.

### 4.1. miRNA: Introduction

MicroRNAs are currently under investigation in various cancer entities for their diagnostic as well as prognostic and predictive roles [79]. MiRNAs are small, approximately 22 nucleotides long, non-coding single-stranded RNAs that regulate gene expression at a posttranscriptional level. The human genome may encode for more than 1000 microRNAs and approximately 60% of human genes are regulated by microRNAs. Besides other functions, they are known to be involved in tumor evolution including regulation of angiogenesis and development of treatment resistance [80,81,82]. Therefore, their respective roles as potential diagnostic and predictive biomarkers have been evaluated: In PC a distinct microRNA expression profile compared to benign lesions has been observe [10,11,83,84]. Distinct miRNA expression profiles correlate to stages of malignant pancreatic disease and hold potentials as biomarkers [85] There is an existing medical need of biomarkers for early diagnosis. CA19-9 is the only prognostic serum-based tumor marker approved in PC; however, it comes with several limitations such as a moderate sensitivity and specificity (estimated around 79% and 82%, respectively). Tumor markers do not always accurately reflect the disease burden; e.g., PC patients with Lewis blood antigen A do not express CA19-9, and false positive results are often seen with the co-existence of inflammation or cholestasis e.g., in case of biliary obstruction [86].

Normally, oncogenes and tumor suppressor genes are regulated at an optimal activation/inhibition equilibrium. If downregulation of a specific miRNA increases the activity of a corresponding oncogene, this is identified as a tumor suppressor miRNA. On the other hand, if upregulation, it will result in a continuous inhibition of the target tumor suppressor gene. An increase in the activity of an oncogene target of an miRNA after knockdown of that miRNA suggests that the miRNA in question acts as a tumor suppressor and conversely overexpression of that same miRNA would lead to increased inhibition of that locus An imbalance can result in the loss of controlling specific tumor formation pathways and can contribute to the development of malignencies [87].

### 4.2. miRNA in Precursor Lesions and Diagnoses

The development of PC is a multistage process of genetic mutations resulting in histological and morphological abnormalities within the ductal cells and the acinar cells of the pancreas [88,89]. These lesions have potential to transform into pre-neoplastic lesions known as pancreatic intraepithelial neoplasia (PanIN).

MiRNA expression profiles are promising as non-invasive diagnostic markers, as they can be obtained easily from peripheral blood, saliva, urine or feces in order to detect above mentioned pre-neoplastic lesions [90]:

Measuring the expression of only 20 to 32 miRNAs could help clinicians in discriminating between healthy, inflamed and cancerous pancreatic tissue [12]. This hypothesis is supported by the fact that upregulation (e.g., miR-21, 221 424-5p, 27a, 4295) as well as downregulation (e.g., miR-124, 203, 150, 218) of miRNAs have been reported to play significant roles during initiation on the one hand and progression on the other hand of pancreatic cancer. Yu et al. investigated the miRNA expression profile in PanIN lesions and reported the aberrant expression of 35 miRNAs. Among those, miR-196b surfaced as a potential biomarker in identifying PanIN-3 lesions [91].

Another microarray analysis study compared blood samples of PC patients and healthy individuals and reported that a distinct miRNA expression profile (miR-22, miR-642b, 10, 752 12 of 22 and miR-885-5p) identified early PC [92].

Furthermore, early stage K-RAS mutations observed in PanIN lesions can directly affect the levels of specific miRNAs as investigated by Humeau et al. [93]. An upregulation of miR-205, miR-200, and miR-21 was detected in early adenocarcinoma lesions in a K-RAS(G12D) mouse model, where miRNA production could be measured in pathological and nonpathological ducts [94].

Increased levels of miR-155 and miR-210 in the serum of pancreatic cancer patient have been reported in various studies [95,96] suggesting a potential role as biomarkers for the diagnosis of early pancreatic neoplasia [95].

Li et al. evaluated a total of 735 miRNAs in the serum of pancreatic cancer patient. This analysis led to the identification of miR-1290 as a promising biomarker [97]. Furthermore, it is reported that miR-1290 displays higher sensitivity (81%) and specificity (80%) when compared to healthy control groups [97,98].

Wang et al. demonstrated that aberrant expression of miR-21, miR-155, miR-196a and miR-210 in plasma can easily distinguish PC patients from healthy controls [99]. Higher than normal levels of miR-210, miR-192 and miR-18a in the serum of PC patients may also be exploited as diagnostic markers [96,100,101].

Another approach in the field of biomarker research is combining various biomarkers or combining them with tumor markers in order to enhance the sensitivity and specificity. Recently, it was reported that the combination of CA 19.9 with miR-16 and miR-196a allows to distinguish between PC patients and healthy controls [102,103]. Similarly, when the expression profile of miR-27a-3p was coupled with CA 19.9, PC patient and healthy controls could be differentiated with a sensitivity of 85.3% and specificity of 81.61% [103,104].

These findings are of utmost clinical relevance, because histological clarification of pancreatic lesions is often challenging, particularly if only small tissue can be obtained by endoscopic ultrasound-assisted fine-needle biopsies. However, no currently existing miRNA panel is endorsed by any guidelines for clinical use to assess response to therapy.

### 4.3. miRNA and Therapy Response

Beyond the diagnostic value of miRNA expression profiles, they might also play a role in the prediction of chemoresistance as well as responsiveness to systemic therapy. MiRNA could help clinicians to choose a combination of various therapies in order to overcome therapeutic resistance.

The predictive value of miRNAs for the response to a therapy with gemcitabine was extensively shown in vitro and in vivo. An overexpression of miR-21 and miR-10b and a downregulation of miR-34a has previously been linked to worse survival under gemcitabine chemotherapy [105,106,107,108,109]. In addition, miRNA-320c has been reported to be of predictive significance for a response to gemcitabine [110].

Furthermore, an irregular expression of miRNA was found in a gemcitabine-resistant cell line, including downregulation of miRNA-200b, miRNA-200c, let-7b, let-7c, let-7d and let-7e in gemcitabine-resistant cells [111]. Similarly, miRNA-33a is also downregulated in gemcitabine-resistant cells, and upon the restoration of normal levels, gemcitabine sensitivity could be restored [112].

In a recent meta-analysis Royam et al. studied a total of 48 miRNAs and reported a downregulation of 23 and upregulation of 25 miRNAs [113]. In particular, nine upregulated miRNAs (15b, 17-5p, 21, 155, 181c, 203, 221,320c and 1246) exhibited chemotherapeutic resistance and six upregulated miRNAs (21, 33a, 138-5p,509-5p, 1207 and 1243) exhibited chemotherapeutic sensitivity. In contrast, nine downregulated miRNAs (7, 100, 124, 210, 200c, 205, 220b, 374b-5p and 497) exhibited chemotherapeutic resistance and nine downregulated miRNAs (101, 101-3p, 153, 203, 205-5p, 494, 506, 3656, let-7a) exhibited chemotherapeutic sensitivity.

This above-mentioned meta-analysis included studies using gemcitabine, 5-FU, capecitabine, and erlotinib. The pooled hazard-ratio (HR) value for OS was 1.603; (95% confidence interval (CI) 1.2–2.143; *p*-value: 0.01), with the subgroup analysis for miR-21 showing a HR for resistance of 2.061; 95% CI 1.195–3.556; *p*-value: 0.09.

Altered expressions of several miRNAs including miR-21-5p, miR-10b-5p, and miR-34a-5p have previously been linked to a worse response to gemcitabine [105,106,107,108,109]. Our group, however, could show that high and low expressions of these three miRNAs have no influence on the outcome of treatment with FOLFIRINOX regarding PFS and OS, in contrast to treatment with gemcitabine [114].

Another study reported by Meijer et al. demonstrated that a decline in plasma miR-181a-5p levels after five to six cycles of FOLFIRINOX was associated with better prognosis [115]. This association was not observed in a second cohort of patients treated with gemcitabine plus nab-paclitaxel. In-vitro analyses detected an increased sensitivity of PC cells lines to oxaliplatin when miR-181a-5p was inhibited. However, to our knowledge there are no miRNA data currently available on response prediction of novel chemotherapy regimens such as nab-paclitaxel or nano-liposomal irinotecan.

A summary of all above-mentioned micro-RNA results can be found in Appendix A.

## Figures and Tables

**Figure 1 biomolecules-11-01469-f001:**
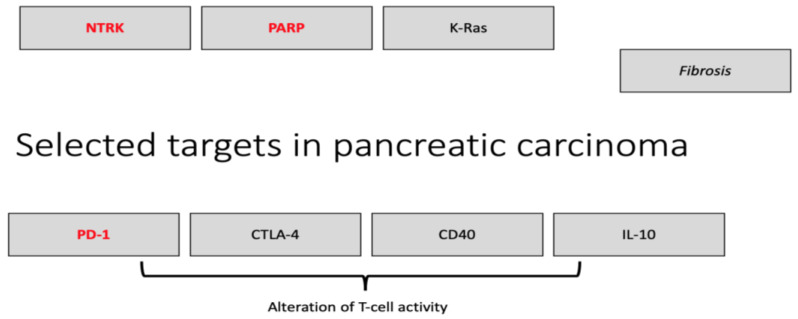
Overview of selected therapeutic targets in locally advanced/metastatic pancreatic carcinoma. Red indicates that therapies have already been approved for this tumor entity.

## Data Availability

Not applicable.

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
