# Peer review of "Recent Advances in Pancreatic Cancer: Novel Prognostic Biomarkers and Targeted Therapy—A Review of the Literature"

_biomolecules, 2021, doi:10.3390/biom11101469_

Round 1

Reviewer 1 Report

The manuscript entitled “Recent Advances in Pancreatic Cancer” provides an overall of the past and recent approaches to targeting pancreatic cancers, with their limitations and benefits.

The manuscript needs to be improved in the English language, there are many typos and I suggest a deep overall re-reading.

Also, I suggest some part needs to be implemented:

  1. Given the importance of the TGF-B signaling in pancreatic cancer tumorigenicity (please cite the following papers:  doi: 10.1038/s41388-020-1289-1, https://doi.org/10.3390/cancers13040930, https://www.nature.com/articles/s41598-020-66908-8), the authors must mention the most recent therapies aimed at inhibiting the TGF-B secretion from pancreatic stellate cells (PSC) and cancer associate fibroblast (CAF) as well as approaches focused to delete these populations, although their success remains controversial.
  2. The manuscript must contain a specific paragraph on strategies aimed at inhibiting the ECM Production (e.g. collagens, laminins, proteoglycans, etc..)
  3. I suggest adding a table in which the authors list all the clinical trials that they cite in the manuscript

Author Response

Reviewer #1:

We thank the reviewer for the statement that this review provides an overall of the past and recent approaches to targeting pancreatic cancers, with their limitations and benefits. The manuscript needs to be improved in the English language, there are many typos and I suggest a deep overall re-reading.

Also, I suggest some part needs to be implemented:

  1. Given the importance of the TGF-B signaling in pancreatic cancer tumorigenicity (please cite the following papers:  doi: 10.1038/s41388-020-1289-1, https://doi.org/10.3390/cancers13040930, https://www.nature.com/articles/s41598-020-66908-8), the authors must mention the most recent therapies aimed at inhibiting the TGF-B secretion from pancreatic stellate cells (PSC) and cancer associate fibroblast (CAF) as well as approaches focused to delete these populations, although their success remains controversial.

We thank the reviewer for this suggestion. A paragraph discussing the role of TGF-beta has been added to topic number five.

  1. The manuscript must contain a specific paragraph on strategies aimed at inhibiting the ECM Production (e.g. collagens, laminins, proteoglycans, etc..)

We have extended the paragraph discussing fibrosis as a therapeutic target by adding the vitamin A as a target and slightly modifying the discussion on vitamin D as a target, as both are known to decrease ECM production.

  1. I suggest adding a table in which the authors list all the clinical trials that they cite in the manuscript
    • We are happy to implement this suggestion. We added a table listing the trials described in our manuscript. Fur furhter deatails please see Supplemental Table 1

Reviewer 2 Report

Although research on pancreatic cancer continues to improve the understanding of the disease progression, resistance development and treatment modalities, Pancreatic cancer is still, sadly, a virtual death sentence with a five-year survival rate which only recently entered double digits this year. It is therefore imperative to assess the current advances in the care and biology of this deadly cancer to provide a curated resource of the current treatment modalities (both in clinical trials, in current approved use and in development) as well as the potential biomarkers that will enable its early detection to aid better outcome for patients after early detection.

The introduction is a little scant at the beginning of the review, and while I appreciate the subsequent introductive paragraphs for the various subtopics, I feel a more comprehensive introduction that will provide cohesion to the review and give a brief overview of the sections that the work has been divided into is necessary. Alternatively, a conclusion statement that explains the import of the current advancements and gaps in understanding and how they impact the care of patients/the field of pancreatic cancer research would also be helpful.

There are a few minor errors that can be addressed as well.

Line 36 : …given the side effects of (instead of by) erlortinib…

Line 61:…because of BRCA1/2 mutation, are sensitive (instead of sensible)

Line 73: …display an increase in OS from 14 to… (is this statement in reference to the use of veliparib in combination with standard of care therapeutics or platinum-based treatments only as a secondline treatment for patients?) please clarify…

Please consider… A phase III cinical trial to determine the most efficacious sequence of administration of PARP inhibitors to enhance OS is required… to replace last sentence starting on line 77

Line 81: … Approximately/Around 95 % of PC patients harbor an activating mutation….

Line 115: …rephrase? ,…master regulator of the network evolving around PC cells… not sure if it is meant to say a  master regulator of gene networks for the evolution of PC cells in mouse models of PC…

Line 125: Overview over of selected therapeutic targets….

Line 197 …, a prove proof of concept study

Lines 244-247: If downregulation of a specific miRNA…target suppressor gene…  please rephrase to make intent a little clearer… implication: an increase in the activity of an oncogene target of an miRNA after knockdown of that miRNA suggests that the miRNA in question acts as a tumor suppressor and conversely overexpression of that same miRNA would lead to increased inhibition of that locus? Please use an example from one of the  miRNAs you have previously mentioned above for clarity of possible.

Several murine models of pancreatic cancer have suggested that both acinar and ductal cells cans serve as the origin of neoplastic lesions in the pancreas. Please reference PMID: 29069604; PMID: 19208745 and others.

Please include a table of the various treatment modalities and the clinical trials they have been associated with a highlight of those approved for current use.

A table of the miRNAs associated with the different stages of disease/ survival outcomes/ resistance signatures would be useful as well.

Author Response

Reviewer #2:

We thank the reviewer for this valid point that it is imperative to assess the current advances in the care and biology of this deadly cancer to provide a curated resource of the current treatment as well as the potential biomarkers that will enable its early detection to aid better outcome for patients after early detection.

The introduction is a little scant at the beginning of the review, and while I appreciate the subsequent introductive paragraphs for the various subtopics

            We added more information to our introduction in order to provide cohesion to    the review. Changes are highlighted accordingly

There are a few minor errors that can be addressed as well.

Line 36 : …we change of (instead of by)

Line 61:…we corrected sensitive (instead of sensible)

Line 73: …display an increase in OS from 14 to… (is this statement in reference to the use of veliparib in combination with standard of care therapeutics or platinum-based treatments only as a secondline treatment for patients?) please clarify…

This is a valuable comment. We clarified, that this was a single-arm trial (veliparip and standard therapy), where patients harboring a mutation in DNA damage repair genes had a median OS of 19 months, compared to 14 months in patients with a wild-type status in these genes.

Please consider… A phase III cinical trial to determine the most efficacious sequence of administration of PARP inhibitors to enhance OS is required… to replace last sentence starting on line 77

Line 81: … we added Approximately 95 % of PC patients harbor an activating mutation….

Line 115: …rephrase? ,…master regulator of the network evolving around PC cells… not sure if it is meant to say a  master regulator of gene networks for the evolution of PC cells in mouse models of PC…

We thank the reviewer for this comment and have rewritten this sentence to adequately describe the role of the vitamin D receptor on the stroma.

Line 125: Overview over of selected therapeutic targets: we corrected accordingly.

Line 197 …, a prove proof of concept study: done

Lines 244-247: If downregulation of a specific miRNA…target suppressor gene…  please rephrase to make intent a little clearer… implication:

 an increase in the activity of an oncogene target of an miRNA after knockdown of that miRNA suggests that the miRNA in question acts as a tumor suppressor and conversely overexpression of that same miRNA would lead to increased inhibition of that locus? Please use an example from one of the  miRNAs you have previously mentioned above for clarity of possible.

            We re-wrote this sentence

Several murine models of pancreatic cancer have suggested that both acinar and ductal cells cans serve as the origin of neoplastic lesions in the pancreas. Please reference PMID: 29069604; PMID: 19208745 and others.

The discussion on the origin of PC has been modified to include acinar cells as well and the two suggested papers have been gladly inserted.

Please include a table of the various treatment modalities and the clinical trials they have been associated with a highlight of those approved for current use.

We thank the reviewer for this suggestion and have included a table that summerizes the clinical trials mentioned in the text. (Suplemental Table 1)

A table of the miRNAs associated with the different stages of disease/ survival outcomes/ resistance signatures would be useful as well.

            We added and new Table (please see Supplemental Table 2) highlighting the role of miRNAs on the outcome